# Exploring the Structural Variability of Dynamic Biological Complexes by Single-Particle Cryo-Electron Microscopy

**DOI:** 10.3390/mi14010118

**Published:** 2022-12-31

**Authors:** Megan C. DiIorio, Arkadiusz W. Kulczyk

**Affiliations:** 1Institute for Quantitative Biomedicine, Rutgers University, 174 Frelinghuysen Road, Piscataway, NJ 08854, USA; 2Department of Biochemistry and Microbiology, Rutgers University, 75 Lipman Drive, New Brunswick, NJ 08901, USA

**Keywords:** single-particle cryo-electron microscopy, cryo-EM, heterogeneity, single-particle analysis, SPA, deep learning algorithms

## Abstract

Biological macromolecules and assemblies precisely rearrange their atomic 3D structures to execute cellular functions. Understanding the mechanisms by which these molecular machines operate requires insight into the ensemble of structural states they occupy during the functional cycle. Single-particle cryo-electron microscopy (cryo-EM) has become the preferred method to provide near-atomic resolution, structural information about dynamic biological macromolecules elusive to other structure determination methods. Recent advances in cryo-EM methodology have allowed structural biologists not only to probe the structural intermediates of biochemical reactions, but also to resolve different compositional and conformational states present within the same dataset. This article reviews newly developed sample preparation and single-particle analysis (SPA) techniques for high-resolution structure determination of intrinsically dynamic and heterogeneous samples, shedding light upon the intricate mechanisms employed by molecular machines and helping to guide drug discovery efforts.

## 1. Introduction

Proteins are dynamic biomolecules that often occupy an ensemble of different conformations to carry out their cellular function. Therefore, understanding the mechanism by which these flexible molecular machines operate requires three-dimensional insight at multiple functional states. Several biophysical techniques aim to resolve high-resolution 3D structures of biological macromolecules. To-date, X-ray crystallography has been an extremely powerful tool for structural biologists to produce atomic models, accounting for over 170,000 depositions into the Protein Data Bank (PDB) [1]. However, successful application of this technique poses several challenges that restrict its use for certain sample types. Biomolecular crystallography requires the formation of well-ordered 3D crystals, making it ill-suited to study proteins and complexes with flexible, unstable, or disordered regions. In such cases, crystallographers may promote crystallization by providing a stabilizing substrate [2] or by genetically removing flexible regions [3,4], but these approaches may yield structures that do not represent biologically relevant states of the protein. Furthermore, even after biochemical purification, certain complexes may exist in a variety of conformational and compositional states, inhibiting the formation of a well-ordered crystal lattice. In addition, samples must be purified to sufficient purity and milligram concentrations for crystallization trials, which often poses a formidable challenge.

Single-particle cryo-EM has become the preferred method for high-resolution structure determination for a variety of protein complexes and biomolecular assemblies. This technique illuminates structural information of molecular machines in their hydrated state without the need for crystallization or large amounts of purified protein. As little as 0.1 milligrams may be sufficient for structure determination [5]. The past decade has seen a sharp rise in the use of single-particle cryo-EM largely attributed to the introduction of novel direct electron detectors (DED), improvements in microscope optics, computer hardware, and image processing software. Deemed the “resolution revolution”, [6] these advances have enabled routine structure determination at high-resolution (reviewed in [7]) and have even made it possible to reach atomic resolution in some cases [8,9]. Moreover, cryo-EM has emerged as an extremely useful technique for the structure determination of large and dynamic biological macromolecules that adopt different structural states [10,11,12].

Heterogeneity is often key to understanding the mechanism of action of biological assemblies, as a single static structure cannot describe the intricate molecular motions employed by proteins during their functional cycle. Cryo-EM is uniquely poised to study protein and nucleic acid complexes with a variety of compositional and conformational states; the nature of sample preparation preserves the native solution protein structure and captures molecules at random orientations and potentially in varying conformations. Furthermore, a variety of computational approaches have been developed in the past decade to resolve different structural states present within the same sample [13,14,15,16,17,18,19]. Nonetheless, these techniques present a number of challenges that make determination of multiple, high-resolution structures far from routine; many are computationally expensive, and their applicability is largely dependent on the particular dataset and prior knowledge of heterogeneity [17,18,19]. Furthermore, newly developed techniques designed to address the continuous flexibility of biomolecules lack validation methods. This review serves to describe the recent advances in cryo-EM methodology that provide high-resolution structural information of heterogeneous samples and, in turn, shed light upon the dynamic mechanisms of action employed by biological macromolecules. We will first describe a typical cryo-EM workflow and overview the origins of sample heterogeneity. Next, we will review the methodology used to analyze biochemical reactions at any given timepoint, also known as time-resolved cryo-EM (trEM). We will then briefly overview the process of 3D reconstruction and methods designed to resolve discrete sample heterogeneity, followed by a discussion of masked classification and refinement approaches. Lastly, we will describe recently developed reconstruction methods, including those utilizing deep learning algorithms, to resolve continuous conformational changes of flexible biological macromolecules.

## 2. The Cryo-EM Workflow

After biochemical purification of the sample, the cryo-EM workflow begins with rapid plunge-freezing of the specimen. Suitable grids for EM contain uniformly distributed particles at a sufficient concentration suspended in a thin layer of vitreous ice, ideally just nanometers thicker than the largest dimension of the molecule of interest. Grid preparation has remained largely unchanged since the development of the rapid plunge-freezing technique proposed 40 years ago by Jacques Dubochet and collaborators [20]. While semi-automated plunge freezers are now routinely used for vitrification, including the Vitrobot [21,22,23], Leica EM-GP [24], Gatan Cryo-plunge (Gatan Inc., Pleasanton, CA, USA), and others, their operation still requires meticulous manual manipulation by skilled users. Furthermore, as several freezing variables are difficult to accurately control, including humidity, blotting force, and tweezer positioning, ice quality can vary greatly between grids.

To prepare a cryo-EM specimen, a few microliters of the sample are first carefully pipetted onto an EM grid made hydrophilic by prior glow-discharging or treatment of the grid with O_2_ plasma. Then, nearly 99.9% of the sample is removed by blotting with filter paper, immediately followed by rapid plunging of the grid into a bath of liquid ethane. Liquid ethane is preferred over other cryogens (e.g., liquid nitrogen) because its low melting point (−188 °C) and high heat capacity allow for fast grid freezing without the formation of ice crystals. In many cases, such a procedure produces cryo-EM grids containing a thin and uniform layer of vitreous ice (<100 nm) suitable for high-resolution imaging. However, the procedure may lead to a number of artifacts. Blotting drastically increases the surface area-to-volume ratio of the sample. Consequently, this process increases the probability of particle collisions with the interface formed between the hydrophobic air and hydrophilic aqueous solution, known as the air-water interface (AWI) [25]. Though the chemistry of the AWI is not relatively well-understood, air exposure nonuniformly disrupts the hydrogen bond network at the AWI, creating a hydrophobic surface that attracts apolar molecules, including hydrophobic residues or small hydrophobic patches of biomolecules [25]. Protein adsorption to the AWI can introduce numerous artifacts to the sample, including partial particle denaturation, as well as the introduction of preferred orientations and compositional heterogeneity. As particles can adhere to the AWI in milliseconds [25], minimizing the time between blotting and vitrification is crucial to produce high-quality grids. Several approaches have been developed to modify protein behavior on grids in order to sequester particles away from the AWI, including the use of graphene support layers [26,27] and functionalized films [28,29,30]. Furthermore, developments in vitrification technology, such as microfluidic spraying [31,32], pin-printing [33], and piezoelectric dispensing [34], have been made to reduce the blot-to-vitrification time or cease the need for blotting altogether.

Vitrified samples are visualized in a transmission cryo-electron microscope, where noisy 2D projections of particles are recorded on a detector. Biological macromolecules are mainly composed of low-molecular weight atoms, thus they can be considered weak phase objects that generate very little contrast when in focus. Therefore, to generate image contrast, the objective lens strength is adjusted to defocus the microscope. The resultant defocused images are modified by the contrast transfer function (CTF), the effects of which are compensated for during image processing. To avoid radiation damage to the sample, images are taken with a low dose of electrons and therefore have a low signal-to-noise ratio (SNR). Moreover, electron exposure induces motion within the vitrified specimen, leading to image blurring [35]. The advent of DEDs with high spatial resolution and fast frame acquisition rates has enabled the collection of movies of the same field of view rather than a single image. Currently, most cryo-EM microscopes facilitate data collection through automated data acquisition software [36,37,38,39], with typical data collection occurring over days. These sessions often produce thousands of movies [40] to image enough particle images for high-resolution reconstruction via single-particle analysis (SPA).

Several different software programs are available for SPA [16,41,42,43,44,45,46,47]. Although these software packages follow a similar workflow for image processing (Figure 1), each program employs different algorithms in several steps that yield 3D reconstructions of varying quality and resolution. We have recently developed a robust pipeline combining the programs cryoSPARC [16], RELION [44], and Scipion [42] for high-resolution structure determination applicable to a variety of datasets [48]. In a typical SPA workflow, movie frames are first aligned and averaged to produce micrographs with increased SNR followed by the correction for aberrational errors imposed by the CTF. Particle selection from these micrographs is a nontrivial task, as it can be difficult to identify particles within noisy, low-contrast images. Typically, hundreds of thousands of particle images are needed to achieve a high-resolution reconstruction. Thus, particle selection is performed through semi-automated or fully automated approaches, but the performance of these algorithms varies greatly depending on micrograph quality. Additionally, the results often include false positives, such as noise, radiation-damaged particles, or partial particles that must be filtered in subsequent classification steps. However, it is important to note that 2D and 3D classification can never completely remove “junk” from the dataset. In the popular template-based approach, the user produces templates for automated particle picking by either manually selecting particles from a subset of micrographs or, in the later stages of processing, by using 2D projections from a previously determined 3D structure [49]. Such a method, however, might be subject to biased selection [50]. Recently, several deep learning approaches have been developed for particle picking [51,52,53,54,55], but as most require manual particle selection or user-provided templates for network training [51,52,53], they too are susceptible to bias. Nonetheless, these methods have demonstrated the potential to improve particle selection from heterogeneous datasets [51,52,53,54,55]. Following picking and extraction, particles are sorted into 2D class averages reflective of different specimen views. 2D classification is iterated until the artifacts from picking are removed, including contaminations and radiation-damaged particles. The particles from the final round of 2D classification are then used as input for 3D structure calculations of the initial model or multiple models. The model or models are then refined to produce high-resolution 3D structures. At this stage in SPA, sample heterogeneity can be addressed through a variety of iterative 3D classification and refinement methods detailed in a later section of this review. The resultant refined 3D Coulomb potential maps can be used to build atomic models given sufficient resolution and map quality.

## 3. Structural Heterogeneity in Cryo-EM Samples

Biomolecular complexes are intrinsically dynamic molecular machines that undergo functionally relevant compositional and conformational changes to perform their cellular roles [59]. Even after purification, protein assemblies remain flexible and often have multiple degrees of freedom in solution. Variation in structural states, usually referred to as sample heterogeneity, poses several problems for structural study via cryo-EM. Notably, the presence of heterogeneity in a dataset can severely limit the achievable resolution of a given sample. During 3D reconstruction, projections with the same orientation are aligned and averaged to increase the SNR, illuminating high-resolution details of the target structure such as residue sidechains and ligands. Averaging of regions with structural variability, however, leads to blurring of Coulomb densities, affecting the resolution of the final reconstruction.

Structural variability within a sample can be the product of compositional and/or conformation heterogeneity. Compositional heterogeneity, the presence of different molecular species, may arise in multiple stages of the cryo-EM workflow. Firstly, prior to vitrification, sample components must be purified or extracted in vivo individually or as a complex. Because biochemical purification very rarely results in complete purity, samples usually contain contaminants. These contaminants might either directly interact with the complex of interest or indirectly interfere with its assembly. Furthermore, the mixing of protein complex components often results in the presence of multiple molecular species, each possibly reflecting a different stage of the catalytic cycle the particular protein complex performs under physiological conditions. The molecules may also form different oligomeric species in vitro, and complexes may vary in subunit stoichiometry depending upon the strength and stability of binding interactions between individual components. This type of heterogeneity can often be mitigated with changes to sample preparation procedures. Complex stability can be improved by optimizing buffer conditions (i.e., pH, salt concentration, etc.) and the number of molecular species can be assessed using native polyacrylamide gel electrophoresis (native PAGE). Moreover, subunits can be covalently linked via chemical cross-linking, though it is important to note that cross-linking can give rise to non-physiological structures. Gradient Fixation (GraFix) [60] has demonstrated success in purifying and stabilizing complexes for cryo-EM [61,62] by ultracentrifugation through a density and cross-linker gradient. As mentioned above, compositional heterogeneity may also arise as a result of grid freezing, as particle collision with the AWI formed during sample blotting can cause partial or complete complex dissociation and/or protein denaturation [25].

Conformational heterogeneity, which can be described by the presence of flexible protein components that can adopt more than one structural state, is often much more difficult to resolve. Conformational heterogeneity may be discrete. It may, for instance, represent two different conformational states of the protein complex. In such a case, the two different conformations may likely be sorted out into distinct 3D structures by the process of 3D classification (Figure 1). However, continuous heterogeneity represents the situation in which conformations are too similar to be differentiated. In cases of discrete heterogeneity, each molecule occupies one of multiple distinct structural states in equilibrium, where the states represent local minima in the energy landscape of protein conformations. Cases of discrete conformational heterogeneity may include the binding of a ligand [63,64], the association and dissociation of subunits [65,66], and the opening and closing of channels in biological membranes [67,68]. In cases of continuous conformational changes, one or more flexible regions of the biomolecular assembly adopt a continuum of conformational states, much exceeding the sorting capabilities of the current 3D algorithms. For example, ribosomes can adopt multiple ratcheted states in which both subunits undergo continuous rotations [12]. Currently, the conformational changes detectable by SPA algorithms are largely limited to domain motions that occur on the timescale of milliseconds; motions of rotamer sidechains and flexible loops, however, cannot be readily visualized. Taken together, although cryo-EM analysis may be complicated for biological complexes displaying varying subunit composition that simultaneously occupy different conformational states, it is often the only available method capable of providing structural insight into such dynamic bio-machines.

## 4. Probing Short-Lived Conformational Changes by Time-Resolved Cryo-EM

Proteins perform their functions on a wide timescale, ranging from microseconds to minutes. Their functional variability often involves structural transitions with transient intermediates representing only a low percentage of the population at equilibrium. Despite their importance to understanding the mechanism of action of molecular machines, these short-lived states have remained elusive to traditional structure determination methods because their lifetime is several orders of magnitude shorter than the time required for sample preparation [69]. Time-resolved cryo-EM (trEM) is a rapidly developing branch of cryo-EM used to study transient structural intermediates by stopping a biochemical reaction at fixed time points via vitrification.

To prepare a trEM sample, a reaction is initiated by mixing reactants, incubated for a desired time interval, and stopped by vitrification. For reactions that occur on the time scale of seconds, this procedure may be achieved by simply mixing reaction components in a test tube immediately prior to grid freezing [70,71,72]. However, traditional mixing with a pipette is rate limiting for much faster reactions that require the time resolution of milliseconds. In such cases, trEM depends on the rapid diffusion of molecules between two thin layers of solution for equilibration. Recently, several “fast” mixing methods have been developed, including on-grid mixing, also known as the spraying-mixing method [73,74,75], microfluidic mixing-spraying [32,76], and light pulse delivery [77,78]. On-grid mixing, developed by Berriman and Unwin [73] and reviewed in [69], requires the direct application of the first reactant onto a grid, thinning of the solution via blotting, and spraying aerosol droplets of the second reactant onto the grid just before vitrification (Figure 2A). This approach has been used to successfully determine cryo-EM structures of the open channel form of acetylcholine receptor [67], the conformational changes of myosin attached to actin during the ATPase cycle [74], and vesicle formation [79]. While this method is applicable to probe protein conformational changes in the presence of small molecules or environmental changes, such as pH or temperature, on-grid mixing is inefficient for reactions that involve interactions between multiple large macromolecules [80]. Moreover, because not all regions of the grid will experience mixing, this technique can introduce compositional heterogeneity to the sample. A further limitation to on-grid mixing is the requirement for blotting, which subjects the sample to the multitude of artifacts created by AWI exposure. Several developments have been made in vitrification technology to overcome this hurdle by reducing blotting time or by ceasing the need for blotting altogether, including piezoelectric transducers [34,81,82] and pin-printing and jet cooling [33]. The Spotiton robot [34,82], now commercially available as the Chameleon system [83], is a newly developed vitrification device that employs piezoelectric dispensing to precisely deliver a small droplet of sample, ranging from picoliters to nanoliters, to an EM grid [34]. This process utilizes a piezoelectric material to propagate an acoustic wave towards the liquid sample, exerting enough pressure on the liquid’s surface to eject a droplet from a fine-tipped nozzle [84]. The ejected droplet is delivered to a “self-wicking” grid comprised of nanowires grown on a copper surface. The nanowires substitute for filter blotting by removing excess liquid from the grid to create a thin layer of particle-containing solution [82,85]. Utilizing an on-grid mixing approach, the Spotiton has recently been used to investigate RNA polymerase promoter binding, conformational changes of potassium ion channels, and dynamin constriction during GTP hydrolysis [68]. The VitroJet [33] is another automated freezing device that utilizes pin-printing to deliver sub-nanoliter volumes of sample to the grid, thereby eliminating the need for blotting, but currently does not have the capability for time-resolved studies [76].

Microfluidic mixing-spraying offers a more efficient and controlled approach for trEM studies, as reactants are mixed prior to grid application. In this technique, both reactants are injected to a silicon chip where they are mixed, incubated for a defined time period, and sprayed onto an EM grid (Figure 2B). In one popular variation of this method, the chip contains a T-shaped mixer followed by a four-tandem butterfly mixing element that serves to merge the fluid streams of the reactants, forming a large interface through which reactants can diffuse across [32]. Using this approach, complete mixing can occur in as little as 0.5 ms [32]. The mixed reactants then traverse through the reaction channel, whose chip-dependent length is dictated by the desired reaction time, ranging from 4 to 500 ms [86]. Lastly, the reaction meets compressed nitrogen gas to generate aerosol droplets that are sprayed onto the grid and immediately plunge frozen. The time resolution of microfluidic mixing-spraying is limited by the speed of the mechanical plunging, which can last more than 10 ms [65,87]. As this technique ensures rapid and thorough mixing of both reactants, it does not have the apparent size-limitations imposed by on-grid mixing. Microfluidic mixing-spraying has been applied to study multiple protein–protein interactions, including ribosome subunit association [65,66] and intermediates of translation initiation [88]. Notwithstanding, to achieve desired flow rates (~6 μL/s) and sufficient particle distribution on the grid [69], this approach requires larger volumes of sample at higher concentrations compared to those needed in traditional blotting methods. These factors limit the applicability of microfluidic mixing-spraying for macromolecules that are difficult to purify. Furthermore, the equipment required to facilitate microfluidic mixing-spraying is currently specialized, expensive, and not widely available.

In a recent study, microfluidic mixing-spraying trEM was used to visualize the conformational changes of release-factor activation during translation termination [89]. During translation termination, the newly synthesized protein is released from the ribosome upon encountering a stop codon. The release is facilitated by a class-1 release factor (RF) that hydrolyzes the ester bond between the tRNA and polypeptide chain. The RFs in bacteria, RF1 and RF2, recognize stop codons in the ribosome decoding center (DC) via a stop-codon-reading (SCR) motif and facilitate ester bond hydrolysis in the ribosomal peptidyl transfer center (PTC) through a GGQ motif. The DC and PTC are separated by a distance of 70 Å, but previously reported crystal structures show that the SCR and GGQ motifs of RF1 and RF2 are located just 20 Å apart [90,91]. Fu et al. employed trEM to visualize the conformational changes in RF1 and RF2 immediately after RF-binding to the pre-termination ribosome [89]. They utilized microfluidic mixing-spraying trEM to capture the termination complexes 24 ms and 60 ms after binding (Figure 3). For both the 24 ms and 60 ms reactions, the authors separately injected the purified release complexes and RFs to corresponding microfluidic chips, followed by mixture spraying onto grids and plunge-freezing. Subsequent SPA revealed that at 24 ms, a quarter of the ribosome-bound RF population is in the compact form (Figure 3A,C), with a similar orientation as the previously reported crystal structures [90,91]. At 60 ms, almost the entire population of the ribosome-bound RF is in the extended form, and the polypeptide is still present in the ribosome exit channel (Figure 3B,D). These structures indicate that during the transition from the compact to extended state, the GGQ motif is placed within the PTC and several key DC residues undergo structural rearrangements. The authors also analyzed a reaction incubated for five hours using traditional mixing and vitrification techniques and observed the RF in only the extended state with no density occupying the ribosome exit channel. Based on these trEM experiments, the authors proposed a stepwise, structure-based mechanism for translation termination in which the initial binding state is the pre-accommodated RF-ribosome complex observed at 24 ms, followed by the catalytic accommodated RF-ribosome complex observed at 60 ms. Finally, at a later time point, the peptide is released from the exit channel.

## 5. 3D Classification & Refinement: Approaches to Modeling Discrete Heterogeneity

There are a variety of reconstruction methods used to obtain a 3D density map from 2D particle projections. One traditional approach utilizes the 3D Radon transform (RT), in which RTs of 2D projections are used to recover the 3D RT, whose inverse is the density map of the target structure [92]. Other widely used methods employ the projection-slice theorem [93] to calculate a 3D volume from 2D projections in Fourier space. Cryo-EM data suffers from multiple limitations that make 3D reconstruction a difficult task. Firstly, particles have random and unknown orientations that must be computationally resolved in order to obtain an accurate reconstruction. Moreover, as noted previously, images are collected with a low SNR and thousands to millions of individual particle images with the same orientation must be averaged to achieve sufficient signal. A third obstacle and the focus of this review is the presence of structural heterogeneity in the cryo-EM sample. Datasets with heterogeneity contain 2D projections corresponding to multiple different 3D structures [93]. Classifying projections from different structures is difficult in 2D because one cannot discern with high confidence different conformational states from different orientations. Therefore, heterogeneity is often addressed through 3D classification.

Initial 3D classification methods were reference-based or so-called “supervised” [94]. Conventionally, these methods apply a projection-matching approach [56], in which a similarity measurement, most commonly the cross-correlation coefficient, is used to compare experimental particle projections with 2D projections of one or more 3D reference(s). Classes and projection angles are then assigned to particles given the orientation of the refence yielding the highest similarity measure. If all projection directions are present in the sample, refinement should result in an improved structure in comparison to the reference model. Thus, this process is iterated until the model no longer improves. Choosing an appropriate initial model is crucial to this approach. Because projection-matching algorithms converge to the nearest local minimum, the initial model must be sufficiently close to the true structure to arrive at the correct reconstruction. In the case of a dataset with structural variability, projection-matching can be applied using multiple 3D references, but this approach requires prior knowledge of the heterogeneity present in the sample [94,95].

To circumvent issues described above, multiple alternative classification techniques have been developed that do not require prior knowledge of sample heterogeneity [16,44,57,96,97,98,99], the most utilized of which are “unsupervised” methods based on maximum-likelihood (ML) procedures [5,57]. ML estimation has been used by x-ray crystallographers to refine electron density maps for decades [100] and was first proposed for 2D alignment of EM images by Sigworth in the late 1990’s [101]. ML approaches have since been extended to classify 3D reconstructions [102], known as 3D maximum likelihood (ML3D) [57], and have been implemented in multiple software packages, including Xmipp [41], Frealign [97], cryoSPARC [16], and RELION [44]. Unlike conventional projection matching, the ML approach does not assign each projection a best-fitting orientation. In brief, ML refinement utilizes an expectation-maximization algorithm to find the most likely reconstructions that describe a heterogeneous dataset by iteratively integrating over all possible probabilities of particle orientations and class assignments [103]. ML3D has been applied to solve multiple structures present in the same sample, revealing intermediates of mRNA-tRNA translocation [104] and different rotational states of the human mitochondrial ribosome [105]. Despite its successes, ML methodologies have several shortcomings. The search for all possible 3D volumes is computationally expensive and does not guarantee convergence. Furthermore, ML algorithms can suffer from model bias, especially in cases of datasets with low SNRs, if the initial reference does not accurately describe the heterogeneity in a given dataset [13]. Ab initio methods that utilize a stochastic gradient descent (SGD) algorithm have been implemented in cryoSPARC [16] to alleviate model-bias, but this approach does not converge to a global minimum and thus requires subsequent refinement to yield a high-resolution reconstruction [106]. Moreover, both ML and SGD methods are designed to find discrete conformations and are limited to the discovery of a small number of different structural states, as these approaches require the user to define the number of 3D classes empirically. If too few classes are used, artificial classes with combined features will form and relevant structural states present in the dataset may go unnoticed. On the other hand, the use of too many classes requires excess computational expense and might lead to separation of different views of the same target complex into different classes. Additionally, specimens that undergo continuous molecular movement cannot be accurately described by these methods, as modeling a continuum of structures in a finite number of classes leaves unresolved heterogeneity within each individual class.

## 6. Masking-Based Approaches to Resolve Discrete Structural States and Continuous Flexibility

As noted previously, cryo-EM datasets often contain multiple types of structural heterogeneity. For example, particles may vary in subunit stoichiometry or select regions of the target complex may undergo conformational changes relative to one another. In such cases, masked 3D classification, or focused classification, can be a useful method to sort the multitude of conformers present in the dataset and improve the resolution of flexible regions [14]. In this approach, the user designs and applies a 3D mask to the region of interest in the 3D structure. The mask excludes all other regions of the structure. The particles are then aligned with only the masked area and every iteration of classification is applied to this selected region. The resultant 3D reconstruction excludes all parts of the structure that do not lie within the mask, thereby sorting the data into subsets that vary only in the user-defined area. The same principal can also be applied to 3D refinement, called a focused refinement, to improve the resolution of flexible components [14]. When generating masks for this approach, it is important that the reference volume is low-pass filtered as to not include high-resolution details that may lead to overfitting and thereby overestimation of resolution during gold-standard Fourier shell correlation (FSC) calculations. Additionally, the mask should have smooth edges to prevent overfitting [14].

During masked classification and refinement, particle images are compared with 2D projections of the masked reference. While the projections of the masked 3D reference only contain the target area of interest, the experimental projections contain information about the entire particle, including the densities that lie outside of the masked region. During the comparison of the masked projections and experimental images, the signal from these unmasked densities in the experimental images acts as noise [14,58,107]. The impact of this additional noise on refinement largely depends on the SNR of the images, as well as the size of the entire complex and subunits subjected to masked refinement [12,58]. To resolve this issue and improve the alignment of the target area, a so-called signal subtraction protocol can be applied in which the signal excluded from the masked density is subtracted from the experimental 2D projections [58]. Signal subtraction approaches in cryo-EM were first used to address symmetry mismatches of bacteriophage φ29 [108] and flaviviruses [109], and similar approaches have since been developed in EMAN [110] and RELION [58]. To perform signal subtraction, two masks must be designed, where one encapsulates the regions that will be used for subsequent focused classification or refinement and the other contains the entire complex except for the area that lies within the first mask [14]; projections of the initial map with the second mask applied are subtracted from the experimental images. Because the experimental images are affected by the CTF, projections of this masked density must first be convoluted with the CTF prior to subtraction [14]. After subtracting these CTF-affected projections from the experimental images, focused classification or refinement can be performed; parameters like orientational searching may need adjustment to improve particle alignment [14]. Signal-subtracting and focused refinement have been applied to probe the mobility of human γ-secretase [58], investigate conformational heterogeneity within the individual subunits of GroEL [111], and resolve the binding between SARS-CoV2 variants and the angiotensin-converting enzyme 2 (ACE2) receptor [112] (Figure 4).

These approaches can be further extended to describe the continuous flexibility of a system through a method called multi-body refinement [15,113,114]. This recently developed approach implemented in RELION [15] models flexible components as multiple, rigid bodies, whose preserved structures vary in orientation relative to each other. Multi-body refinement assumes compositional homogeneity of the target complex (i.e., individual molecules have the same subunit stoichiometry) and that motion is resigned to a finite number of two or more bodies that move independently of one another [15]. For instance, when applying this approach to a dataset of the *Plasmodium falciparum* cytoplasmic ribosome [12], Wong et al. observed independent motion in the head and body domains of the ribosome, and thus defined these regions as two separate bodies. Furthermore, multi-body refinement assumes that each body is present in every particle in the dataset with their relative orientation subject to change from particle to particle. First, a consensus refinement is performed to generate an initial map from which the user designs separate masks that contain each independent body. For every iteration of multi-body refinement, 2D projections are generated for each masked body and the signal from the surrounding bodies is subtracted; particles are separately aligned against the densities of each independent body and the relative orientation of all bodies are recorded. In doing so, signal subtraction is improved for each iteration. To visualize the motion within the dataset, principal component analysis (PCA) is performed on the relative orientations of each body to produce movies of 3D volumes that describe the largest variability in the system [15]. Multi-body refinement was first applied to resolve conformational heterogeneity of the tri-snRP complex [114] and has since been used to investigate the dynamics of a multitude of complexes and assemblies, including the conformational dynamics of G protein-coupled receptor (GPCR) and arrestin protein binding [115] and the mechanism by which SARS-CoV-2 Nsp1 binds to the human 40S ribosomal subunit to inhibit translation [116].

It is important to note that each target region for multi-body or focused refinement must be sufficiently large, typically greater than 150 kDa [15], for poses to be accurately assigned, limiting their applicability to complexes with smaller flexible regions. Furthermore, there is currently no standardized method for atomic model building from the multiple, independently refined maps produced by these approaches. Programs such as UCSF Chimera [117], Phenix [118], and Coot [119] offer tools to combine maps generated from multi-body refinement to produce a composite structure. However, artifacts may be present at the interfaces between subunits, as the rigid body assumption no longer holds where interface residues undergo conformational changes [14,15]. For example, multi-body refinement of a pre-catalytic spliceosome shows a chemically unfeasible broken helix connecting the spliceosome core and Sf3b body [15]. In another case, after applying a focused refinement approach to determine a high-resolution structure of the bacterial ribosome, Watson et al. observed degradation of the highest resolution components at the interfaces of maps [120].

## 7. Focused Classification and Multi-Body Refinement of Ribosomal Complexes

Ribosomes are molecular machines that facilitate protein synthesis by converting messenger RNA (mRNA) into chains of amino acids. These large macromolecules are composed of up to 80 different proteins and three to four RNA molecules, forming a small and large subunit that both undergo conformational changes to facilitate the functional cycle of translation. The ribosome is an ideal target for cryo-EM analysis, as its large size (2.7 MDa in *E. coli*) generates sufficient contrast needed for the accurate alignment of particle projections. Furthermore, because ribosomes undergo numerous conformational changes throughout translation initiation, elongation, and termination, there are often multiple structural states present in the same sample. Due to the aforementioned computational advances to address sample heterogeneity, the past few years have seen a surge in ribosome structures at near-atomic resolution (reviewed in [121]). In particular, focused refinement and classification techniques and multi-body refinement have enabled many high-resolution reconstructions that facilitate the study of transcription-translation coupling [122,123], as well as ribosomal interactions with translation factors [124,125,126,127,128,129]. Furthermore, these techniques have allowed for the identification of chemical modifications [120,130,131,132], analysis of potential drug interactions with protein side chains and nucleic acids [12,63,64,130,133,134,135,136], and the visualization of ribosomal complexes at high-resolution in situ [137,138].

In an interesting example, Khawaja et al. utilized focused 3D classification and multi-body refinement to calculate two high-resolution structures of distinct pre-initiation states of mitochondrial translation and investigate the conformation changes that occur during complex assembly [139]. Translation initiation in the human mitochondria requires the assembly of the mitochondrial ribosome with mRNA and mitochondrial initiation factors 2 (mtIF2) and 3 (mtIF3). To elucidate the mechanism of complex assembly, Khwaja et al. applied cryo-EM and SPA to study a sample containing the mitochondrial ribosome small subunit (mtSSU), mtIF2, and mtIF3. After initial pre-processing and preliminary 2D and 3D classification of the dataset, Khawaja et al. performed focused classification and signal subtraction on the mtIF3 binding site to isolate particles containing mtIF3. To further separate complexes containing only mtIF3 from those bound by both mtIF2 and mtIF3, the authors applied focused 3D classification and signal subtraction on the mtIF2 binding site (Figure 5A). This workflow resulted in two maps representing mitochondrial preinitiation steps 1 and 2 (mtPIC-1, mtPIC-2) at 3.0 Å and 3.1 Å resolution, respectively (Figure 5A). Khawaja et al. also performed masked local refinement on the head core, body core, tail, and ms39 regions of both maps to improve local resolution of these regions. Initial rounds of 3D classification revealed multiple, similar conformations of the mtSSU head indicative of a continuous conformational change. Thus, the authors used multi-body refinement to elucidate the relative orientations of the mtSSU head and body, applying soft masks to the head and body regions. Multi-body refinement and subsequent PCA revealed a head-swiveling motion (Figure 5B). Analysis of 3D volumes corresponding to different head swiveling states suggested that the positioning of mS37, a 13.5 kDa protein bound to the mRNA exit channel, serves to restrict head swiveling and allows for the accommodation of mtIF2 in the second step of translation initiation.

## 8. Approaches to Modeling Continuous Heterogeneity

As detailed above, many developments have been made to successfully reconstruct a small number of discrete conformations present within the same cryo-EM dataset. Solving high-resolution structures of systems that exhibit a continuum of functional states, however, poses challenges and is currently an area of rapid development (Table 1). Several PCA-based techniques [140,141,142] have been proposed to model continuous heterogeneity, including 3D variability analysis (3DVA) [17] implemented in the popular software package cryoSPARC [16]. 3DVA can be used to resolve both continuous flexibility and discrete heterogeneity without size limitations or the need for an underlying model for motion. Based upon the work by Tagare et al. [142], 3DVA is a linear subspace model that uses an expectation-maximization algorithm for probabilistic PCA, with the goal of finding the top principal components, or eigenvectors, that correspond to the molecular variability within the dataset. Together, the principal components describe the linear subspace comprised of the conformers present in the dataset. 3DVA requires a previous consensus reconstruction to generate poses of the experimental projections, as the algorithm assumes that the changes in conformation are small enough such that the correct projection directions can be assigned using a single consensus structure. Consequently, it is recommended to perform prior 3D classification in cases where large compositional heterogeneity is known or expected in the dataset [17]. The algorithm then determines the position of each particle along the trajectory of each principal component and represents the particle positions as reaction coordinate plots. The molecular motion within the dataset is visualized through “movies” of 3D volumes along a given principal component. However, because 3DVA is a linear interpolation along eigen volumes, motion is visualized by the appearance and disappearance of densities. 3DVA has been used to resolve large molecular motions of the pre-catalytic spliceosome [17], ratcheting motions of the ribosome [17], and the flexibility of small complexes, including a 53-kDa region of a GPCR complex [17] and the Stx2a-ribosomal P-stalk complex [143]. Furthermore, 3DVA has been instrumental in investigating the dynamics of SARS-CoV-2 proteins [144,145,146,147,148,149].

Machine learning approaches have also gained traction in the past few years for describing the continuous flexibility of various complexes and assemblies, a subject of another review in this volume entitled “Novel artificial intelligence-based approaches for ab initio structure determination and atomic model building for cryo-electron microscopy” [150]. Recently developed methods include ManifoldEM [18,151,152,153], CryoDRGN [19], CryoGAN [154,155], e2gmm [156] and 3DFlex [157]. ManifoldEM [18] is a nonlinear manifold embedding method used to describe flexible variation of a system across its energy landscape. As in 3DVA, this approach requires a consensus reconstruction to determine particle orientations [86]. In ManifoldEM, the 2D images with similar projection directions are combined to form a conformational manifold that is represented in the low-dimensional space of principal components. The dimensionality of the space is dictated by the degrees of freedom of the specimen. A 3D movie is produced to show the conformational change along any given projection directions. The manifolds corresponding to each projection direction must be stitched together to form a single map that describes the spectrum of continuous conformational changes of the system [158,159]. The consolidated map is used to generate a free energy landscape of the system. For any selected point in the map, conventional reconstruction methods can be applied to yield a 3D volume. ManifoldEM has been used to investigate conformational variability in a large dataset of the 80S yeast ribosome [153], as well as the SARS-CoV-2 spike protein [160]. Similar manifold learning strategies have also been employed to analyze synthetic datasets [161]. ManifoldEM uses raw images as input and ceases the need for preliminary classification or prior knowledge of the number of different structural states present within the dataset. However, it should be noted that this method requires a large number of datasets as well as the fine-tuning of parameters at several steps for successful execution.

**Table 1 micromachines-14-00118-t001:** Recently developed methods for heterogeneous reconstruction of biomolecular structures displaying continuous conformational changes.

Method	Advantages	Disadvantages	Reference
ManifoldEM	Generates free-energy landscape of the system	Fine tuning of hyperspace parameters	Frank & Ourmazd, 2016 [18]
AlphaCryo4D	Applicable to small proteins	Requires large dataset Oversamples conformational space	Wu et al., 2022 [162]
CryoDRGN	New version [159] does not require initial model or pose information Resolves discrete and continuous conformations	Long training time Empirical optimization of latent space	Zhong et al., 2021 [19]
CryoGAN	Does not require initial model or pose information Resolves discrete and continuous conformations	Limited resolution of reconstructions	Gupta et al., 2021 [154]
e2gmm	Reduces parameters needed to represent particles Intuitive interpretation by Gaussian parameters	Requires large amount of GPU memory Limited to small proteins for high resolution	Chen & Ludtke, 2021 [156]
3DVA	Resolves discrete and continuous motion No fine-tuning of parameters Applicable to small proteins	Not applicable to systems with nonlinear geometry Artifact of appearing/disappearing densities	Punjani et al., 2021 [17]
3DFlex	Models motion directly instead of 3D volume	Auto-decoder is computationally expensive	Punjani et al., 2022 [157]
Multi-body refinement	Automated implementation in RELION [44] Improves subdomain resolution	Interfaces between bodies poorly resolved Size limitation for densities < 150 kDa	Nakane et al., 2018 [15]

CryoDRGN (Deep Reconstructing Generative Networks) [19] is another popular machine-learning method that employs two deep-learning neural networks to resolve continuous heterogeneity in a cryo-EM dataset. CryoDRGN uses particle images and poses to train an image-encoder-volume-decoder architecture based on a spatial variational autoencoder (VAE). The algorithm encodes the 2D particle projections into the low-dimensional latent space, learns the structural variability within the system, and based on the Fourier projection-slice theorem, decodes slices of corresponding 3D volumes [19]. Similar to ManifoldEM, the user-defined dimensionality of the continuous manifold in the latent space describes the heterogeneity of the system. As a consequence of the deep-learning neural network architecture, the ability to resolve continuous conformational changes directly depends upon the parameters defining the latent space. Thus, the latent space is subject to empirical fine-tuning [19]. The most recent version of CryoDRGN has implemented an ab initio algorithm to determine particle orientations during image encoding [163]. In contrast, the original version of the program relies on a previous consensus refinement to solve for poses. CryoDRGN presents the resultant reconstructions as the parameters of the neural network that can be visualized as a voxel array of points. CryoDRGN has been used to probe molecular motions of the spliceosome [19], the 80S ribosome [19], and non-ribosomal peptide synthetases [164]. A recent study utilized CryoDRGN to reveal a tilting motion of the radial spokes of dynein motors [165].

While the above-described approaches show great promise in resolving molecular motions of large macromolecular machines, they rely on expert knowledge and experimental methods to attribute biological significance to the calculated motion and lack standardized and robust methods for validation. Traditional reconstruction methods may be performed to verify the presence of structural states obtained from these methods [19], or multiple techniques may be used in conjunction to validate the observed movements [165,166]. For example, the same “twisting” and “hinging” motions observed in human recognition complex structures by multi-body refinement were also shown in the top principal components of 3DVA (Figure 6) [166]. Furthermore, attributing biological significance to motions learned from cryo-EM data still requires confirmation by biophysical and biochemical experiments (e.g., NMR or other spectroscopy methods, single-molecule analysis, etc.). Additionally, there is the risk of over-interpreting motion within the dataset that has no biological meaning.

## 9. Conclusions

Cryo-EM is a rapidly developing tool for high-resolution structure determination with the capability to resolve the structural variability of biological macromolecules and assemblies. Developments in sample preparation and vitrification methodologies have enabled the structural study of biochemical reactions at specific time points, allowing structural biologists to visualize elusive structural intermediates and gain insight into the function of dynamic assemblies. However, improvements are still needed to facilitate widespread adoption of trEM, namely efforts to minimize the amount of sample required for study and the development of affordable and reliable devices for sample preparation. Furthermore, advances in computational methods have enabled users to extract multiple structural states from a cryo-EM sample in silico. There are now many software packages available to obtain high-resolution structures, as well as multiple different classification and refinement strategies that can be applied to study heterogeneous samples; however, their applicability largely depends on the specifics of the dataset. For example, masking-based approaches have demonstrated the ability to resolve multiple discrete structural states, as well as improve the resolution of flexible densities. While major strides have been made to address continuous motion, including deep learning-based approaches, these methods are still very much in their infancy and lack standardized validation.

## Figures and Tables

**Figure 1 micromachines-14-00118-f001:**
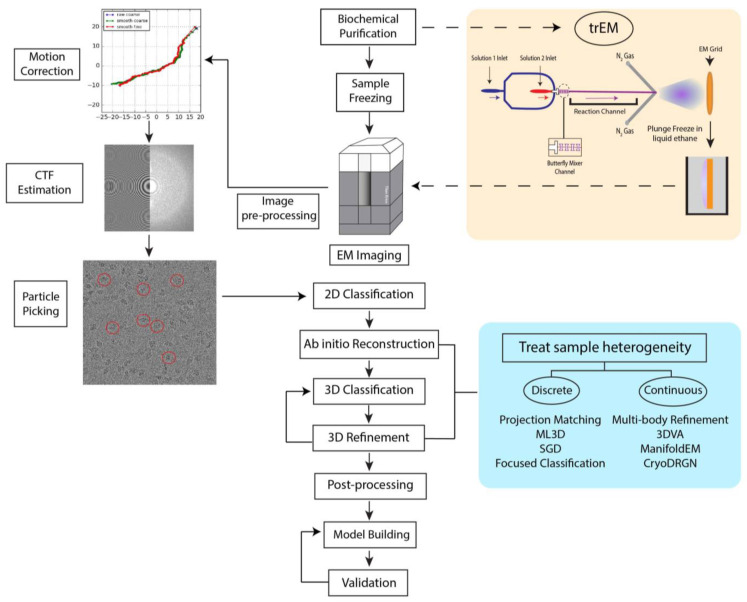
Schematic representation of a typical cryo-EM processing workflow. After biochemical purification, samples are rapidly plunge-frozen in liquid ethane and imaged in an electron microscope. In time-resolved cryo-EM (trEM) (orange box), biochemical reactions are frozen at different time points to elucidate structures of reaction intermediates. The schematic in the orange box depicts a microfluidic mixing-spraying method that utilizes a microfluidic chip to mix, incubate, and spray reactants onto a cryo-EM grid prior to plunge-freezing. The data is acquired with a cryo-electron microscope. The resultant data is processed in multiple stages, including image pre-processing, 2D and 3D classification, post-processing, and model building. Image pre-processing steps (left) include motion correction, CTF estimation, and particle picking, in which particles are selected for downstream processing. The selected particles are indicated by red circles. Following the 2D classification of particle images, various strategies can be employed to resolve sample heterogeneity. These strategies are outlined in the blue box. Multiple 3D reconstruction approaches are available to resolve discrete heterogeneity, including projection matching [56], 3D maximum likelihood (ML3D) [57], ab initio modeling using stochastic gradient descent (SGD) [16], and focused classification [58]. Recently, several methods have also been developed to address continuous heterogeneity, including multi-body refinement [15], 3D variability analysis (3DVA) [17], manifold embedding (ManifoldEM) [18], and CryoDRGN (Deep Reconstructing Generative Networks) [19]. Image post-processing, model building, and structure validation complete the workflow.

**Figure 2 micromachines-14-00118-f002:**
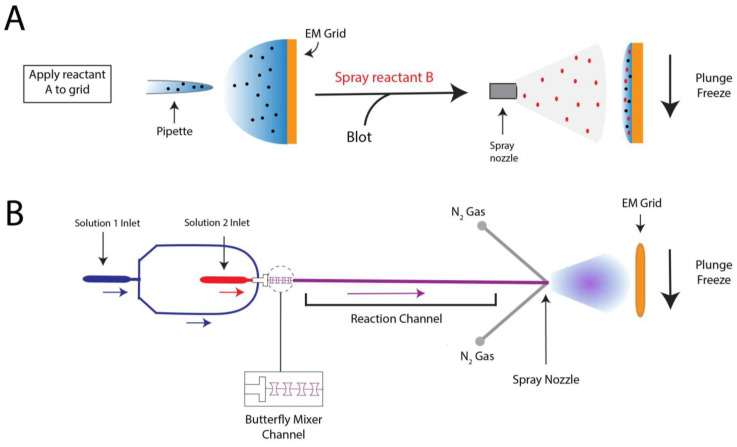
Sample mixing techniques for trEM. (**A**) Schematic of on-grid mixing, in which the first reactant is pipetted onto the cryo-EM grid, followed by blotting of the grid and spraying of the second reactant. The grid is then plunge-frozen. (**B**) In microfluidic mixing-spraying, the reactants (1 and 2) are injected into separate inlets of the microfluidic chip and mixed via a T-shaped mixer followed by four-tandem butterfly geometries [32]. The mixed reactants then traverse the reaction channel, whose length varies depending upon the desired reaction time. After meeting compressed nitrogen gas, the mixture is sprayed to a cryo-EM grid and plunge frozen. Colored arrows indicate the direction of the solution flow.

**Figure 3 micromachines-14-00118-f003:**
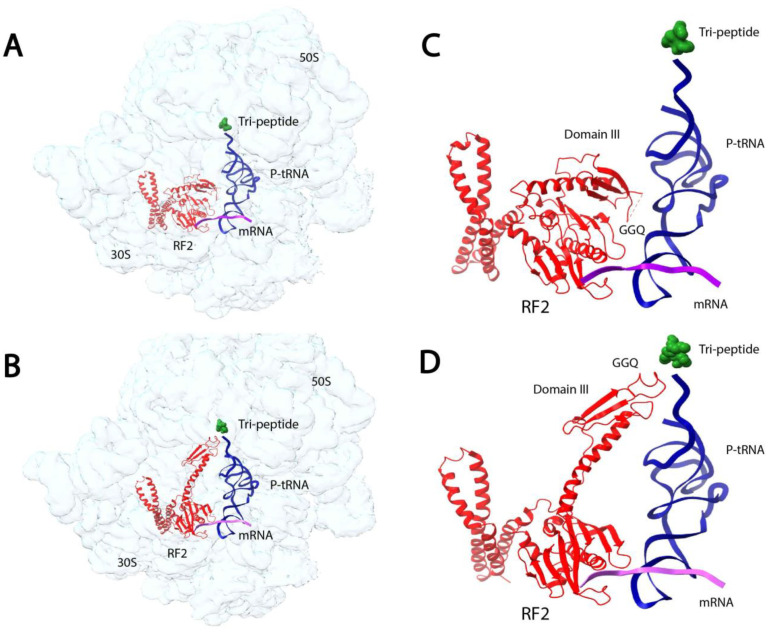
TrEM reveals structural intermediates of release-factor activation during translation termination [89]. (**A**,**B**) Cryo-EM structures of the *E. coli* 70S ribosome bound to RF2 in compact (**A**) and extended (**B**) conformations. After incubating the reaction for 24 ms, 25% of the ribosome-bound RF population was in the compact form. At 60 ms, the entirety of the observed population was in the extended form. (**C**,**D**) Ribosome-bound RF2 in compact (**C**) and extended (**D**) states show the location of the GGQ motif and domain III. RF2 is shown in red, tRNA in blue, mRNA in purple, and the tripeptide in green. The compact state map and model can be accessed with the accession codes EMD-20188 and PDB ID:6OST, respectively. The extended state map and model can be accessed with the accession codes EMD-20193 and PDB ID:6OT3, respectively.

**Figure 4 micromachines-14-00118-f004:**
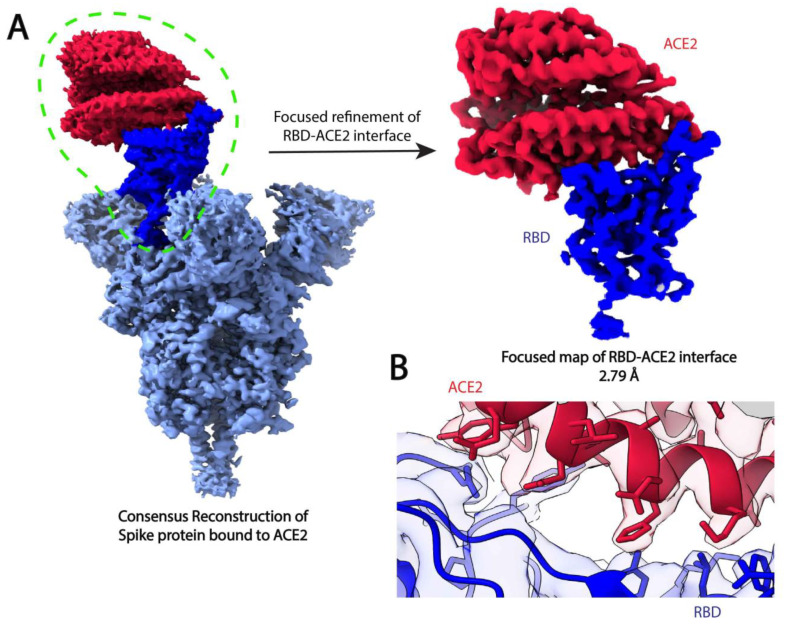
Focused refinement of the SARS-CoV-2 spike protein receptor binding domain (RBD) and ACE2 improves the resolution of the RBD-ACE2 interface [112]. (**A**) Consensus refinement of the SARS-CoV-2 D614G mutant spike protein in complex with the human ACE2 ectodomain resulted in a reconstruction with global resolution of 2.66 Å, but the RBD-ACE2 interface had a local resolution of 6.2 Å. Focused refinement of the RBD (dark blue) and ACE2 (red) substantially improved the resolution of this region to 2.79 Å and allowed for (**B**) the visualization of sidechain rotamer arrangements at the interface fitted with atomic coordinates (PDB ID: 7SXX). The global refinement and focused refinement maps can be accessed with the accession numbers EMD-25509 and EMD-25510, respectively.

**Figure 5 micromachines-14-00118-f005:**
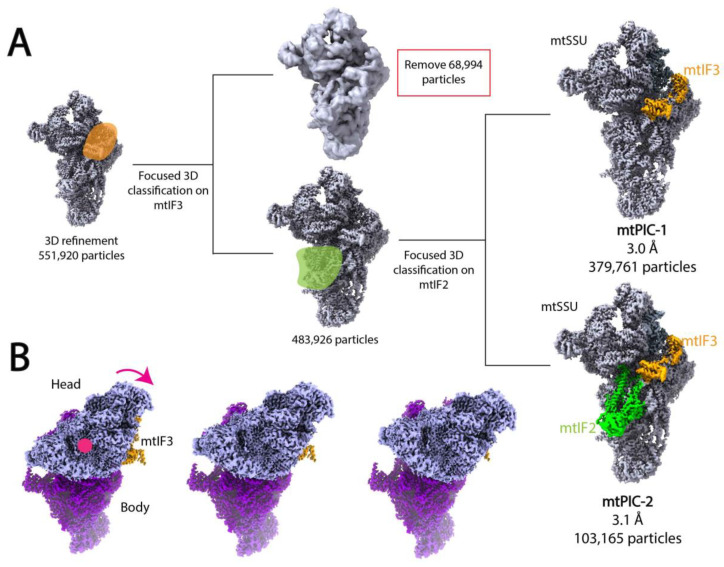
Focused classification elucidates two high-resolution intermediates of human mitochondrial translation and multi-body refinement reveals mtSSU head-swiveling motion during complex assembly [139]. (**A**) Workflow of focused classification. After 3D refinement, Khwaja et al. performed two rounds of focused refinement and signal subtraction masking the mtIF3 and mtIF2 regions to separate mtIF3 only (mtPIC-1) and mtIF3-mtIF2 bound (mtPIC-2) complexes. mtSSU is shown in gray, mtIF3 and its mask in orange, and mtIF2 and its mask in green. Maps of mtPIC-1 and mtPIC-2 can be accessed with accession numbers EMD-10021 and EMD-10022, respectively. (**B**) Multi-body refinement and subsequent PCA of mtPIC-1 revealed a head swiveling motion emanating from the rotation of rRNA h28. Depicted maps only serve to represent motion and were not experimentally determined. The mtSSU head is shown in light purple, body in purple, and mtIF3 in orange. The pink arrow indicates the direction of motion and pink dot represents the axis of rotation.

**Figure 6 micromachines-14-00118-f006:**
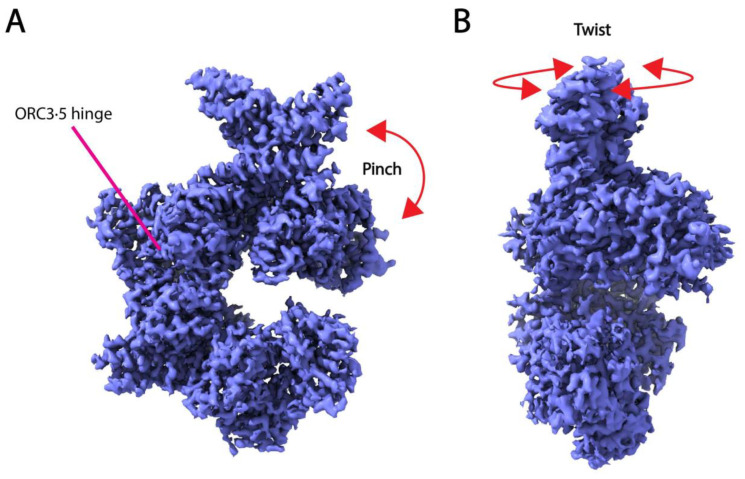
Twisting (**A**) and pinching (**B**) movements of the open human recognition complex (ORC) were revealed by PCA employed by multi-body refinement in RELION and 3DVA in cryoSPARC [166]. Both motions emanate from a hinge at the interface of the ORC3 and ORC5 subunits. The cryo-EM map shown in the figure is the map of the open ORC, (EMD-22417).

## Data Availability

Not applicable.

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
