# Peer review of "Exploring the Structural Variability of Dynamic Biological Complexes by Single-Particle Cryo-Electron Microscopy"

_micromachines, 2022, doi:10.3390/mi14010118_

Round 1

Reviewer 1 Report

The work by Dilorio and Kulczyk is a quite nice and comprenhensive review on the topics of cryoEM and its use to explore structural variability on macromolecules. I have just very few comments:

-Introduction. Perhaps too optimistic, comment also on limitations from the very beginning

-Figure 1: Do not use so early "trEM", but spell out time resolved EM

-On section 5, and although it is extremely common in the field, I would place its roots on on Radom transform and not so much on Fourier Space arguments 

-CryoGAN was expanded to several classes by Unser group

-For all methods, you should not necessarily attribute biological meaning to any mathematical entity (PCA, manifolds...)

-In ManifoldEM a major task is the stitching of manifolds. Please complete with the newer references from Frank group

-On CryoSparc 3DVA, authors might want to also refer to Hemant previous work on the same topic 

Author Response

We would like to thank the Reviewer for taking the time to review the manuscript. We appreciate the comments and constructive criticism, which gave us the opportunity to revise and improve the manuscript. The concerns raised by the Reviewer are addressed below.

 Introduction. Perhaps too optimistic, comment also on limitations from the very beginning

Limitations of the workflow are now discussed in the introduction. Please see lines 64 – 69:

“a variety of computational approaches have been developed in the past decade to resolve different structural states present within the same sample [13-19]. Nonetheless, these techniques present a number of challenges that make determination of multiple, high-resolution structures far from routine; many are computationally expensive, and their applicability is largely dependent on the particular dataset and prior knowledge of heterogeneity [17-19]. Furthermore, newly developed techniques designed to address the continuous flexibility of biomolecules lack validation methods.”

Figure 1: Do not use so early "trEM", but spell out time resolved EM

A revised figure caption spells out “time-resolved cryo-EM”.

On section 5, and although it is extremely common in the field, I would place its roots on Radom transform and not so much on Fourier Space arguments 

The discussion of Fourier Space arguments has been limited in the section 5 (previous lines: 360 – 364). We have included the following section concerned with the RT in the manuscript (lines 364 -368):

“There are a variety of reconstruction methods used to obtain a 3D density map from 2D particle projections. One traditional approach utilizes the 3D Radon transform (RT), in which RTs of 2D projections are used to recover the 3D RT, whose inverse is the density map of the target structure [92]. Other widely-used methods employ the projection-slice theorem [93] to calculate a 3D volume from 2D projections in Fourier space.”

CryoGAN was expanded to several classes by Unser group

The table 1 now contains reference to Multi-CryoGAN. The following text has been removed (CryoGAN row): “Limited to one reconstruction of a single conformation”.

For all methods, you should not necessarily attribute biological meaning to any mathematical entity (PCA, manifolds...)

We thank the Reviewer for this comment. We have removed the following statements:

-“Furthermore, as described below, attributing biological interpretation to the manifold is a potentially difficult and time-consuming task currently lacking automation”; (formally lines: 595 – 597)

-“Difficult to attribute biological interpretation to manifold”; (table 1).

In addition, the following statement was added in lines 702 -703: “there is the risk of over-interpreting calculated motion within the dataset that has no biological function”.

In ManifoldEM a major task is the stitching of manifolds. Please complete with the newer references from Frank group

We apologize for this important omission. The following articles from the Frank group are now cited in the revised manuscript:

  1. Maji, S.; Liao, H.; Dashti, A.; Mashayekhi, G.; Ourmazd, A.; Frank, J. Propagation of Conformational Coordinates Across Angular Space in Mapping the Continuum of States from Cryo-EM Data by Manifold Embedding. J. Chem. Inf. Model. 2020, 60, 2484-2491.
  2. Seitz, E.; Acosta-Reyes, F.; Maji, S.; Schawnder, P.; Frank, J. Optimization of ManifoldEM Informed by Ground Truth. IEEE Trans. Comput. Imaging. 2022, 8, 462-478.

The following text has been added to lines 624 – 627 in the revised manuscript:

“A 3D movie is produced to show the conformational change along any given projection directions. The manifolds corresponding to each projection direction must be stitched together to form a single map that describes the spectrum of continuous conformational changes of the system [158,159]. The consolidated map is used to generate a free energy landscape of the system.”

On CryoSparc 3DVA, authors might want to also refer to Hemant previous work on the same topic 

The discussion concerned with CryoSparc 3DVA has been modified, and it now contains the following statement in lines 594 – 597:

“Based upon the work by Tagare et al. [142], 3DVA is a linear subspace model that uses an expectation-maximization algorithm for probabilistic PCA, with the goal of finding the top principal components, or eigenvectors that correspond to the molecular variability within the dataset.”

Reviewer 2 Report

The articles Exploring the Structural Variability of Dynamic Biological Complexes by Single-particle cryo-electron microscopy is a review that focuses on the recent developments in cryo-EM for detecting and studying conformational changes in macromolecular complexes. The review describes time resolved cryo-electron microscopy, the instrumentation needed to perform the studies, the computation analysis involved in determining conformational (discrete or continuous) in macromolecules, and some of the pitfalls associated with the process. The manuscript is well written. Some suggestions for improvements include:

1.    Lines 45-46: "near-native" is a bold statement. The conditions used for SPA are significantly different than conditions within the cell. The cellular milieu is quite viscous, and interactions with neighboring molecules may have an affect on the structure and function of the micromole of interest. While these points do not diminish the impact of structures determined by cryo-EM, calling purified protein “near-native” is incorrect and misleading. Admittedly, all techniques have limitations. The novel reader would benefit from knowing this.

2.    Lines 89-91: Please provide a sentence for the novice reader, and X-ray crystallographers, why liquid ethane is preferred over liquid nitrogen. For the older crystallographer, can liquid propane be used?

3.    Lines 93-95: It may be a good idea to indicate why the air water interface is believed to be hydrophobic. This idea is thrown around the cryo-EM community as it’s common knowledge, but I fear it may not be. The chemical explanation would provide more emphasis on why structural chemistry, underlying structural biology, is so important in understanding how the world works.  

4.    Lines 142-148: "bad" particles are never completely removed during 2D or 3D classification. Maybe inserting a sentence here would help the novice microscopist appreciate this.

5.    Lines 213-215: It may be helpful to indicate how large of a conformational change is detectable by cryo-EM. For example, changes in rotamer side-chains are not readily visualized, nor are loop motions; however, domain motions are more likely to be observed.  May also be helpful to identify the timescale of these motions. Cryo-EM is likely to observe msec motion rather than micro or nanosec motion.

6.     

Author Response

We would like to thank the Reviewer for taking the time to review the manuscript. We appreciate the comments and constructive criticism, which gave us the opportunity to revise and improve the manuscript. The concerns raised by the Reviewer are addressed below.

  1. Lines 45-46: "near-native" is a bold statement. The conditions used for SPA are significantly different than conditions within the cell. The cellular milieu is quite viscous, and interactions with neighboring molecules may have an affect on the structure and function of the micromole of interest. While these points do not diminish the impact of structures determined by cryo-EM, calling purified protein “near-native” is incorrect and misleading. Admittedly, all techniques have limitations. The novel reader would benefit from knowing this.

We thank the Reviewer for making this important point. The phrase “near-native” has been replaced with “hydrated state” in the following sentence (lines: 44 - 47):

 “This technique illuminates structural information of molecular machines in their hydrated state without the need for crystallization or large amounts of purified protein.”

  1. Lines 89-91: Please provide a sentence for the novice reader, and X-ray crystallographers, why liquid ethane is preferred over liquid nitrogen. For the older crystallographer, can liquid propane be used?

A following sentence has been added in lines 97-99:

“Liquid ethane is preferred over other cryogens (e.g. liquid nitrogen), because its low melting point (-188°C) and high heat capacity allow for fast grid freezing without the formation of ice crystals.”

  1. Lines 93-95: It may be a good idea to indicate why the air water interface is believed to be hydrophobic. This idea is thrown around the cryo-EM community as it’s common knowledge, but I fear it may not be. The chemical explanation would provide more emphasis on why structural chemistry, underlying structural biology, is so important in understanding how the world works.

We thank the Reviewer for this comment, and agree that understanding of chemical basis underlaying the hydrophobic nature of the air water interface appear to be far from common knowledge. Thus, we have added the following text in lines 104 – 110:

“this process increases the probability of particle collisions with the interface formed between the hydrophobic air and hydrophilic aqueous solution, known as the air-water interface (AWI) [25]. Though the chemistry of the AWI is not relatively well-understood, air exposure nonuniformally disrupts the hydrogen bond network at the AWI, creating a hydrophobic surface that attracts apolar molecules, including hydrophobic residues or small hydrophobic patches of biomolecules [25].”

  1. Lines 142-148: "bad" particles are never completely removed during 2D or 3D classification. Maybe inserting a sentence here would help the novice microscopist appreciate this.

We have added the following text in lines 147 – 150:

“Additionally, the results often include false positives, such as noise, radiation-damaged particles, or partial particles that must be filtered in subsequent classification steps. However, it is important to note that 2D and 3D classification can never completely remove “junk” from the dataset.”

  1. Lines 213-215: It may be helpful to indicate how large of a conformational change is detectable by cryo-EM. For example, changes in rotamer side-chains are not readily visualized, nor are loop motions; however, domain motions are more likely to be observed.  May also be helpful to identify the timescale of these motions. Cryo-EM is likely to observe msec motion rather than micro or nanosec motion.

Prompt by the Reviewer’s comment, we have added the following text in lines 240-243:

“Currently, the conformational changes detectable by SPA algorithms are largely limited to domain motions that occur on the timescale of milliseconds; motions of rotamer sidechains and flexible loops, however, cannot be readily visualized.”